# Clinical Application of the Computed-Tomography-Based Three-Dimensional Models in the Surgical Planning and Intraoperative Navigation of Ureteropelvic Junction Obstruction Complicated with Renal Calculi

**DOI:** 10.3390/medicina58121769

**Published:** 2022-11-30

**Authors:** Shengwei Xiong, Mengmeng Zheng, Chunlin Liu, Kunlin Yang, Zhihua Li, Lei Zhang, Ye Tian, Liqun Zhou, Xuesong Li

**Affiliations:** 1Department of Urology, Peking University First Hospital, Institute of Urology, Peking University, National Urological Cancer Centre, Beijing 100034, China; 2Department of Urology, Beijing Friendship Hospital, Capital Medical University, Beijing 100050, China; 3Department of Urology, Fu Xing Hospital, Capital Medical University, Beijing 100045, China

**Keywords:** three-dimensional computed tomography model, ureteropelvic junction obstruction, renal calculi, stone extraction, pyeloplasty

## Abstract

*Background and Objectives:* To clinically validate the computed tomography (CT)-based three-dimension (3D) model for treatment planning and intraoperative navigation of ureteropelvic junction obstruction (UPJO) complicated with renal calculi. *Materials and Methods:* We retrospectively collected the data of 26 patients with UPJO and renal calculi who were surgically treated in our institution from January 2019 to December 2021. Before surgery, 3D models based on preoperative CT scans were constructed in all patients. Additionally, the patients were divided into three groups according to the results of 3D models, distinguished by different treatment of renal calculi, that is, left untreated (1), pyelolithotomy (2), and endoscopic lithotomy (3). The quantitative analysis of renal calculi parameters, and perioperative and follow-up data were compared. *Results:* The mean number of involved renal calyces (*p* = 0.041), and the mean maximum cross-sectional area (*p* = 0.036) of renal stones were statistically different among the three groups. There were no significant differences among the three groups in the mean operative time, mean estimated blood loss, mean pararenal draining time, and mean hospital stay. The intraoperative stone clearance rates were 100% (11/11) and 77.8% (7/9) in group 2 and group 3, respectively. The trends of increased estimated glomerular filtration rate and decreased creatinine on the average levels after surgery were observed, although these changes were not statistically significant. At a mean follow-up of 19.4 ± 6.4 months, the overall surgical success rate of the UPJO was 96.2% (25/26), and the overall success rate of renal calculi removal was 80.8% (21/26). Renal stones in 66.7% (4/6) of patients in group 1 spontaneously passed out. *Conclusions:* Preoperative 3D CT models have exact clinical value in the surgical planning and intraoperative navigation of UPJO patients complicated with renal calculi.

## 1. Introduction

Ureteropelvic junction obstruction (UPJO) is characterized by decreased flow of urine down the ureter and increased fluid pressure inside the kidney. Additionally, a delayed washout of crystalline aggregates may promote crystal agglomeration, nucleation, and final stone formation. It is reported that the incidence of UPJO with ipsilateral renal calculi is approximately 16–30% [1,2,3], and this coexistence might propose a management dilemma for urologists [4].

A detailed understanding of stones and their anatomical relationships prior to surgery is required to optimize the stone clearance rate and minimize complications. However, ultrasonography, X-ray, and computed tomography (CT) urography provide the images simply in a two-dimensional level. A recently developed three-dimensional (3D) reconstruction technique based on preoperative CT scans provides stereoscopic and lifelike anatomical understanding for surgeons, and allows full surgical planning, from intraoperative orientation to lesions removal, in many fields including renal tumors and urinary lithiasis [5,6]. Prior studies had shown the benefits of 3D reconstruction in minimizing the risks of percutaneous nephrolithotomy and achieving higher intraoperative stone-free rates for patients with complex renal calculi [5], but no study exists evaluating the efficiency of 3D reconstruction technique as a preoperative planning tool for patients with UPJO and renal calculi.

In this retrospective study, individualized surgical planning and grouping for included patients with UPJO and renal calculi according to the 3D CT reconstruction models were developed preoperatively. We also compared the variables of renal stones and hydronephrosis, and the perioperative data of patients in different groups. In addition, intermediate follow-up outcome data of these patients were presented.

## 2. Methods and Materials

### 2.1. Patient Selection and Management

We retrospectively reviewed the data of patients diagnosed with UPJO at Peking University First Hospital from January 2019 to December 2021, and 26 patients with concomitant renal calculi were recruited in this study. Before surgery, 3D CT models based on preoperative CT scans were performed in all patients. These patients underwent pyeloplasty with or without simultaneous treatment of renal calculi. All surgeries were performed by an experienced surgical team. Intraoperative and postoperative complications were recorded and graded according to the Clavien-Dindo classification. The postoperative follow-up was conducted by the surgical team through outpatient clinic at regular intervals (every 3–6 months) after the operation. We have a standardized follow-up team, and the relevant preoperative and postoperative data were recorded in our multicenter RECUTTER (Reconstruction of Urinary Tract: Technology, Epidemiology and Result) database (http://www.3dmi.com.cn/login/pkufh (accessed on 1 March 2022)). Postoperative renal function was evaluated by monitoring the changes in serum creatinine and estimated glomerular filtration rate (eGFR). Ultrasonography and CT urography or 3D CT reconstruction technology were used as follow-up examinations to evaluate postoperative hydronephrosis and residual stones. Surgery success of UPJO was defined as the alleviation of hydronephrosis and resolution of symptoms. Additionally, surgery success of renal calculi was defined as removal of all stones intraoperatively. This study was approved by the Biomedical Ethics Committee of Peking University First Hospital. Additionally, all individual participants signed informed consents preoperatively, including acceptance to receive 3D CT reconstruction technology.

### 2.2. 3D CT Reconstruction and Stone Evaluation

All patients underwent enhanced CT urography performed on a 64-CT scanner (Philips Brilliance 64-MDCT, Philips, Best, The Netherlands). Then, CT images were processed and stored in the Digital Imaging and Communications in Medicine (DICOM) format. The DICOM files were loaded into the 3D visualization system (IPS system, Yorktal, Shenzhen, China) to construct 3D imaging models by technical engineers. The 3D CT models give insight into the shape and location of lesions. The vasculature, renal parenchyma, renal collecting system, ureters, urinary bladder, and lesions were colored with different colors (Figure 1). The location and extent of stricture of the ureteropelvic junction, the anatomical distribution, renal vascular variation, urinary tract stones, and 3D anatomic relationships for different structure combinations were displayed when different degrees of transparency and rotation were adjusted (Figure 1 and Figure 2). Length and volume measurement tools were used to assess the size of stones and the renal parenchymal and pelvicaliceal volume.

### 2.3. Patient Grouping and Surgical Planning

For the treatment of UPJO, traditional laparoscopic pyeloplasty (TLP) and robot-assisted laparoscopic pyeloplasty (RALP) were alternative. In general, if the surgery is expected to be complicated, RALP is preferable. Of course, the choices of patients and their families also needed to be considered.

The reconstructed 3D models permit precise analyses of the renal stones and UPJO. Prior to the surgery, the surgeon carefully reviewed the results from 3D CT reconstruction models. For patients with small renal calculi located deep in the renal calyx (Figure 2A), only TLP or RALP was performed. For patients with renal calculi of bigger sectional area, deep position and more involved renal calyces (Figure 2B,C), TLP or RALP combined with pyelolithotomy using laparoscopic forceps was performed. For patients with more complicated and larger renal calculi or staghorn calculi (Figure 2D,E), TLP or RALP combined with endoscopic lithotomy was performed. Therefore, the included patients were divided into three groups, group 1 “LP”, group 2 “LP with pyelolithotomy”, and group 3 “LP with endoscopy”, respectively (LP means TLP or RALP), for final quantitative analysis.

### 2.4. Surgical Techniques

All procedures were accomplished under general anesthesia. Importing the 3D CT models into the surgical display screen (Figure 3A) allowed live comparison of anatomy between the surgical view and the 3D models, which realized the intraoperative navigation and localization. Thirteen patients underwent modified TLP in a transperitoneal approach. The pneumoperitoneum was established with transperitoneal subcostal access and trocar site layout was completed using the techniques developed by our team [7]. The dilated pelvis and strictured site of ureteropelvic junction (UPJ) were exposed using standard laparoscopic techniques, and a 1.5 cm oblique incision on the lower pole of the pelvis above the strictured site of UPJ is made. As for patients in group 2, the procedure of pyeloplasty was interrupted, and continued with lithotomy. Renal pelvic and calyceal stones were found using laparoscopic forceps with reference to the 3D CT models (Figure 3B), and then the stones were placed in a specimen bag and removed out. As for patients in group 3, flexible ureteroscope or cystoscope was introduced into the renal pelvis through the initial incision (Figure 3C). Subsequently, the stones were extracted under the guidance of the flexible guiding tube with continuous irrigation (Figure 3D). If the renal calculi were large, impacted, or embedded in the calyces, holmium laser lithotripsy was introduced. After stones are removed, the remaining procedures continued as a modified dismembered pyeloplasty (Figure 4), initially described by our team [8]. The remaining 13 patients underwent RALP in a transperitoneal approach, the specific distribution of trocars and the specific surgical steps were referred to that reported by Hong et al. [9]. The procedures of concomitant treatment of renal calculi of patients who underwent RALP are the same as those of patients who underwent LP as described above.

### 2.5. Statistical Analysis

Statistical analysis was performed using SPSS 24.0 (IBM, Armonk, NY, USA). Continuous data are presented as mean ± standard deviation and were analyzed using the Student’s *t* test. Categorical variables were compared using the chi-square test or Fisher exact test. Two-tailed *p* values < 0.05 were considered statistically significant.

## 3. Results

### 3.1. Characteristics of Patients and Renal Calculi

The demographic characteristics of included patients and quantitative assessment of renal calculi are listed in Table 1. There was no statistically significant difference among the three groups in age, gender, body mass index, eGFR, affected renal parenchymal volume, affected renal hydronephrosis volume, hydronephrosis degree, and the spatial distribution (upper, mid or lower calyces) of renal calculi. The mean number of involved renal calyces (*p* = 0.041), and the maximum cross-sectional area of stones (*p* = 0.036) was statistically different among the three groups. As for pairwise comparison, the mean number of involved renal calyces of group 1 was significantly less than that of group 2 (*p* = 0.046), and the mean maximum cross-sectional area of stones of group 1 was significantly smaller than that of group 2 (*p* = 0.044).

### 3.2. Perioperative and Follow-Up Data

Table 2 presents the detailed perioperative and follow-up results for patients in the three groups. Thirteen patients underwent TLP and the remaining patients with RALP, without conversion to open surgery. Additionally, no statistically significant differences in options of pyeloplasty, either TLP or RALP, were found among the three group (*p* = 0.119). Although the patients in group 1 underwent LP alone, but there was no significant difference in the mean operative time when compared to group 2 and 3, that is 150.2 ± 62.3 min, 153.4 ± 47.2 min, and 136.1 ± 37.9 min, respectively (*p* = 0.716). The mean estimated blood loss, the mean pararenal draining time and the mean hospital stay were no significant differences among three groups (*p* = 0.540, *p* = 0.270, *p* = 0.246, respectively). In group 3, seven patients performed lithotomy using flexible cystoscopy (Figure 3B) and two patients using semi-rigid ureteroscopy. Additionally, two of them underwent additional holmium laser lithotripsy, still it failed to clear the stones completely. The intraoperative stone clearance rates were 100% (11/11) and 77.8% (7/9) in group 2 and group 3, respectively. We collected the patients’ renal function results (eGFR and creatinine) before surgery and 1 day, 3 months, and 12 months after surgery to observe the changing trends. The results are shown in Figure 5. We could observe the trend of increased eGFR and decreased creatinine on the average levels after surgery, although these changes were not statistically significant.

Only one patient in group 3 developed Clavien–Dindo grade 3 complication. Additionally, this patient had a displaced ureteral stent 2 days after surgery, which was subsequently replaced with a new one. The remaining patients did not develop Clavien–Dindo grade 2 or higher complications during perioperative period. The double-J stent was removed successfully via cystoscopy 2–3 months after surgery in all patients. At a mean follow-up of 19.4 (6.4–29.2) months, the follow-up ultrasonography confirmed that the hydronephrosis mitigated or stabilized in all patients without recurrence of obstruction. The overall surgical success rate of the UPJO was 96.2% (25/26), as 1 patient recurred back pain 16 months after surgery due to renal stones. In group 1, one patient underwent ureteroscopic lithotripsy nine months after the surgery because of the enlarged and symptomatic stones, and one patient still had stones without apparent progression over the entire follow-up. Additionally, in group 2, one patient underwent ureteroscopic lithotripsy five months after the surgery because of the recurrent stones. Percutaneous nephrolithotomy was performed 2 months after surgery in the two patients of group 3 whose renal calculi were not completely removed during surgery. Additionally, ultrasonography showed that one patient had small amounts of residual stones in the middle renal calices at the last follow-up.

## 4. Discussion

Recently, 3D reconstruction technology has been increasingly adopted to facilitate the surgeon in better developing surgical planning and navigation. The 3D virtual models are constructed based on preoperative cross-sectional images, including CT and magnetic resonance imaging (MRI). Moreover, these virtual models can be printed by the 3D printers for better preoperative training and preparation. Shin et al. reported that the life-size 3D printing models based on the MRI data provided surgeons with precise knowledge of the location of prostate cancer and its relationships with neurovascular bundles [10]. In addition, Bertolo et al. investigated the role of 3D virtual models in complex renal masses [11]. The authors found that the surgeons changed their indication from radical to partial nephrectomy after viewing the respective 3D virtual models in about 27% of the cases [11].

Concomitant renal calculi are not uncommon in patients with UPJO, with an incidence of 16-30% [4,12]. The key to successful operation of UPJO and ipsilateral stones lies in thorough preoperative evaluation. Compared with prostate cancer and kidney tumors, 3D reconstruction technology may be more suitable for complex renal stone diseases, as the 3D virtual models can accurately represent the interrelationships between the collecting system, stones, and adjacent anatomical structures with the aim of minimizing the risks of lithotomy procedures. Brehmer et al. reported that the 3D CT reconstruction technique optimized the selection of the access route into the renal pelvis, improved stone clearance rate, and reduced surgical complications during percutaneous nephrolithotomy (PCNL) [13].

In this study, the affected renal parenchymal volume and affected renal hydronephrosis volume, the number, maximum cross-sectional area, location distribution and the number of affected calyces of stones were quantitatively analyzed (Table 1). Preoperatively, the surgeon subjectively divided the included patients into three groups based on the quantitative assessment of renal calculi by 3D CT reconstruction models. The three groups were named LP, LP with pyelolithotomy, and LP with endoscopy, respectively. The quantitative analysis of stones showed that the mean number of involved renal calyces (*p* = 0.041), the maximum cross-sectional area of stones (*p* = 0.036) was statistically different among the three groups.

The treatment of renal calculi has developed from open surgery to several minimally invasive options. Extracorporeal shock wave lithotripsy, ureteroscopy, and PCNL have been reported to effectively minimize the morbidity of stones removal, and PCNL is considered as the main treatment for large and staghorn stones [14]. The treatment of renal calculi with concomitant UPJO is complex and intractable, and it remained diverse without a standard protocol. PCNL combined with endopyelotomy may not succeed when UPJO is caused by anterior crossing vascular compression [15]. At present, minimally invasive pyeloplasty (TLP or RALP) in combination with pyelolithotomy, rigid nephroscope, flexible ureteroscope, or cystoscope lithotomy have been reported with satisfactory outcomes [16,17,18,19]. Kadihasanoglu et al. reported that the intraoperative stone clearance rate of TLP combined with pyelolithotomy was 93.3% (26/28), but the operative time was no significant difference (*p* = 0.88) when compared with pyeloplasty without pyelolithotomy [16]. Yin et al. reported that the stone clearance rate of TLP in combination with flexible ureteroscope lithotomy was 100% (16/16), and there was no recurrence of stone and obstruction in the follow-up of average 29.3 months [17].

Koh et al. reported 28% (16/57) renal stones measuring 5 mm or less passed spontaneously, significantly more likely than stones that were larger (*p* = 0.006), and there was no difference in the incidence of passage when the stone located the lower, mid or upper renal calyx [20]. In our study, six patients with concomitant stones in the depth of renal calyx with the mean maximum cross-sectional area of 51.50 mm^2^, underwent pyeloplasty alone in group 1. Urinary ultrasound at the last follow-up showed that there were no renal stones in 66.7% patients (4/6), indicating the stone had passed out. In patients with stones not discharged, one patient had symptomatic stones that needed to be managed, and the remaining two patients had stones without progression. The incidence of spontaneous passage in our study was significantly higher than Koh et al. reported, probably because the obstructive of ureteropelvic junction was relieved and the number of included cases was deficient. Based on the evidence, we hypothesized that relief of obstruction may improve metabolic disorder of affected renal unit, and then facilitate the expulsion of stones and reduce the formation of new stones. Thus, conservative management for asymptomatic small concomitant nephronlithiasis (<50 mm^2^) is viable and reasonable.

To the best of our knowledge, our study presents the first report of using 3D reconstruction technique as a preoperative planning tool for the pyeloplasty with concomitant stone extraction. According to our initial experience, in UPJO patients with small calculi located deep in the renal calyx, if the diameter of the stone is <5 mm or the cross-sectional area is < 50 mm^2^, and the number of involved renal calyces is <2, the renal calculi can be left untreated when performing pyeloplasty for the management of UPJO. Part of such stones may spontaneously pass from the kidney into the ureter since the obstruction has been relieved [20,21]. Regular observation is needed because the stones might become larger gradually and then cause symptoms. As for ipsilateral renal calculi with larger sectional area and more involved renal calices, either concomitant pyelolithotomy or endoscopic lithotomy can be adopted. However, we think that concerning complex stones with large quantity, large sectional area (>200 mm^2^), concomitant lithotomy under endoscopies such as ureteroscopy, cystoscopy, and nephroscopy should be given priority.

There are several limitations in this study, including a small number of cases, single-center design, and short-term follow-up. Thus, additional studies with multicenter data, a larger sample size, and long-term follow-up are required to further validate the efficiency of the 3D CT reconstruction technique for the surgical planning and intraoperative navigation of UPJO patients with ipsilateral renal calculi.

## 5. Conclusions

The usage of the 3D CT reconstruction technique for the preoperative planning and intraoperative navigation of UPJO complicated with renal calculi allows for an increased intraoperative stone clearance rate and reduced complications and recurrence of obstruction and stones.

## Figures and Tables

**Figure 1 medicina-58-01769-f001:**
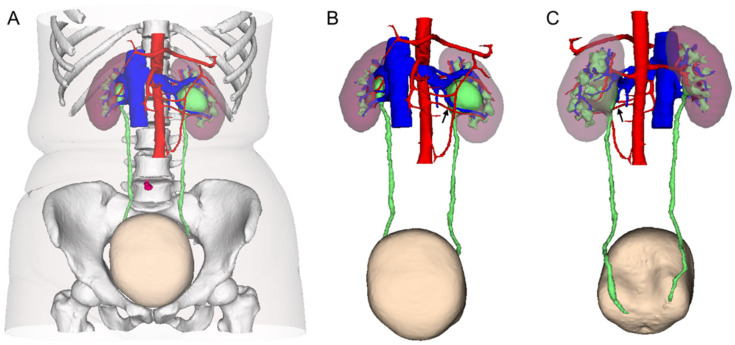
(**A**) Three-dimensional (3D) computerized tomography (CT) reconstruction model showing urinary system and its vascular system, skin, bones, belly button, etc. (**B**) Anterior view of 3D CT reconstruction model showing urinary system and its vascular system only. (**C**) Posterior review of 3D CT reconstruction model showing urinary system and its vascular system only. (Crossing renal artery in the lower pole of the kidney was indicated by black arrow.)

**Figure 2 medicina-58-01769-f002:**
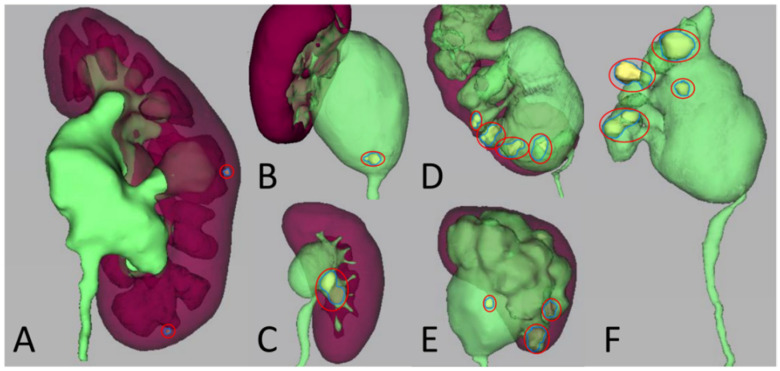
3D CT reconstruction models of renal calculi and adjacent structures such as renal parenchyma, renal pelvis and calyces in different patients. (**A**) stone located in the deep part of renal calyces; (**B**) stone located above the ureteropelvic junction; (**C**) stone located at the openings of multiple renal calyces; (**D**) multiple stones located in different renal calyces; (**E**) multiple stones located in the deep part of renal pelvis and calyces; (**F**) large multiple stones located in the deep part of renal pelvis and multiple renal calyces. (Renal calculi were indicated by red circles.)

**Figure 3 medicina-58-01769-f003:**
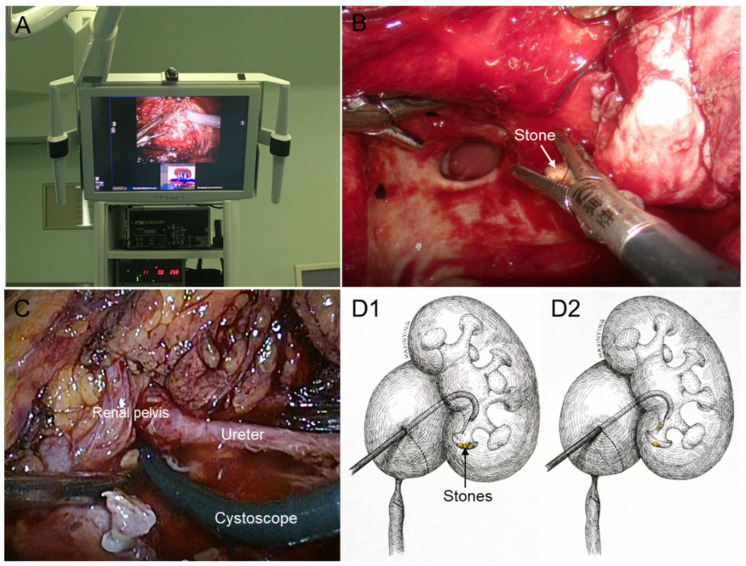
Concomitant procedures of renal calculi removal. (**A**) 3D CT reconstruction model was displayed on the surgical viewing screen for intraoperative navigation and localization; (**B**) the renal calyceal stone was extracted using laparoscopic forceps; (**C**) the flexible cystoscope was introduced into the renal pelvis through an oblique incision on the lower pole of the pelvis above the strictured site of ureteropelvic junction; (**D**) the subrenal calyx stones were extracted by flexible cystoscope (D1, stone exploration; D2, stone extraction).

**Figure 4 medicina-58-01769-f004:**
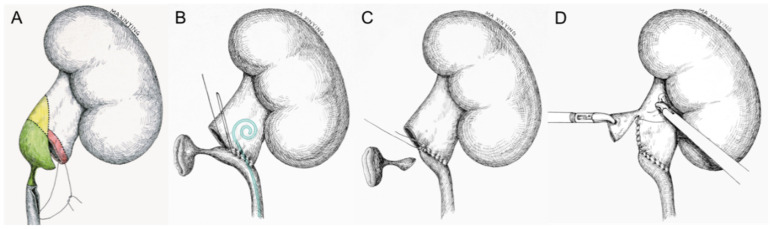
The diagram of modified dismembered pyeloplasty. (**A**) The first stitch is taken between the ureter and the lowest corner of the renal pelvis (the red color representing the area of anastomosis); (**B**) after clipping the connection of the UPJ with the renal pelvis, the anastomosis was performed by running suture starting from the point of first stitch, and then a 7F double-J stent was inserted into the distal ureter; (**C**) the strictured segment of the ureteropelvic junction was removed; (**D**) the redundant renal pelvis was sheared while the margin is continuously sutured, and finally a modified dismembered pyeloplasty was completed.

**Figure 5 medicina-58-01769-f005:**
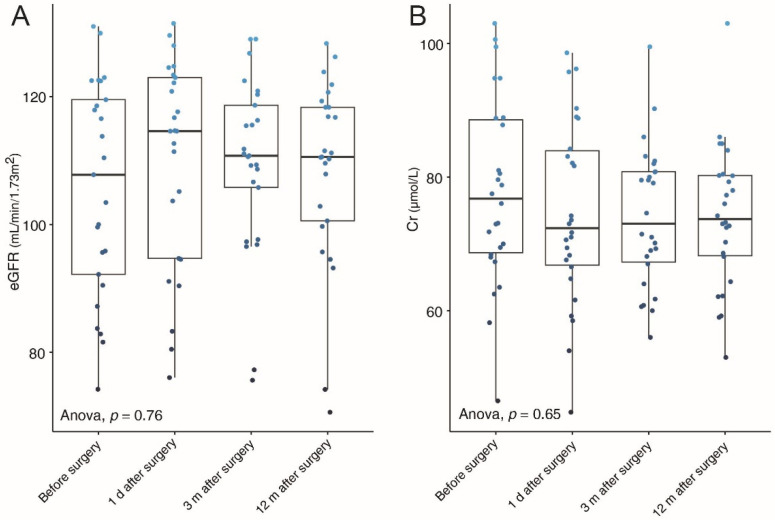
Trends of estimated glomerular filtration rate (eGFR) (**A**) and serum creatinine (Cr) (**B**) from preoperative to postoperative. The blue dots on images represent the values of eGFR or Cr. d = day; m = month.

**Table 1 medicina-58-01769-t001:** Demographics, clinical, and stone assessment data of the patients.

Variables	LP	LP with Pyelolithotomy	LP with Endoscopy	Total	*p* Values
Number of patients, *n*	6	11	9	26	-
Gender, *n*					0.526
Male	3	8	7	18	
Female,	3	3	2	8	
Age, years	36.7 ± 14.2	27.7 ± 8.9	31.9 ± 11.9	31.2 ± 11.7	0.329
BMI, kg/m^2^	22.9 ± 3.5	23.7 ± 2.0	25.1 ± 4.6	24.1 ± 3.6	0.455
Side of UPJO and stones, no. of patients, *n*					0.450
Right	2	2	2	6	
Left	4	7	7	18	
Bilateral	0	0	2	2	
eGFR, mL/min	102.3 ± 18.1	112.6 ± 22.9	110.7 ± 16.7	107.7 ± 15.0	0.590
No. of crossing vessel, *n*	2	2	1	5	0.494
Affected renal parenchymal volume, mL	149.2 ± 52.8	190.7 ± 65.4	156.6 ± 43.1	166.6 ± 54.3	0.360
Affected renal hydronephrosis volume, mL	81.0 ± 26.3	193.8 ± 121.3	126.6 ± 92.4	138.7 ± 144.2	0.413
Hydronephrosis degree, no. of patients, *n*					0.310
Mild	4	7	3	14	
Moderate or severe	2	4	6	12	
Stones, no. of patients, *n*					0.384
Single	4	4	3	11	
Multiple	2	7	6	15	
Distribution of stones, no. of patients, *n*					0.499
Upper calyces	0	1	2	3	
Middle or lower calyces	6	10	7	23	
Involved renal calyces, no. of patients, *n*					0.041 *
≤2,	6	9	4	19	
>3	0	2	5	7	
Maximum cross-sectional area, mm^2^	51.50 ± 29.44	176.30 ± 89.58	347.59 ± 113.98	206.79 ± 89.51	0.036 *

LP, laparoscopic pyeloplasty (including traditional laparoscopic pyeloplasty and robot-assisted laparoscopic pyeloplasty); UPJO, ureteropelvic junction obstruction; eGFR, estimated glomerular filtration rate. * *p* < 0.05.

**Table 2 medicina-58-01769-t002:** Operative data and follow-up data of the patients.

Variables	LP	LP with Pyelolithotomy	LP with Endoscopy	Total	*p* Values
Options of pyeloplasty, no. of patients, *n*					0.119
Traditional laparoscopic	2	4	7	13	
Robot-assisted laparoscopic	4	7	2	13	
Operative time, min,	150.2 ± 62.3	153.4 ± 47.2	136.1 ± 37.9	146.6 ± 46.7	0.716
Estimated blood loss, mL	25.8 ± 26.5	49.1 ± 57.3	32.7 ± 33.2	38.1 ± 43.6	0.540
Intraoperative stone clearance rate, %	0	100% (11/11)	77.8% (7/9)	-	-
Pararenal draining time, day	6.0 ± 5.4	3.4 ± 1.2	3.9 ± 1.2	4.2 ± 3.2	0.270
Length of hospital stay after operation, day	6.7 ± 6.1	4.2 ± 1.6	4.3 ± 1.0	4.8 ± 3.1	0.246
Perioperative complications (>grade 2), *n*	0	0	1	1	-
Follow-up time, month	18.4 ± 6.7	20.1 ± 5.9	21.6 ± 6.3	19.4 ± 6.4	0.648
Success rate of UPJO, *n*/N (%)	5/6 (83.3)	11/11 (100)	9/9 (100)	25/26 (96.2)	0.193
Success rate of renal calculi, *n*/N (%)	4/6 (83.3)	10/11 (90.9)	7/9 (77.8)	21/26 (80.8)	0.592

LP, laparoscopic pyeloplasty (including traditional laparoscopic pyeloplasty and robot-assisted laparoscopic pyeloplasty); UPJO, ureteropelvic junction obstruction.

## Data Availability

Not applicable.

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
