# Peer review of "Clinical Application of the Computed-Tomography-Based Three-Dimensional Models in the Surgical Planning and Intraoperative Navigation of Ureteropelvic Junction Obstruction Complicated with Renal Calculi"

_medicina, 2022, doi:10.3390/medicina58121769_

Round 1

Reviewer 1 Report

This interesting single-center study by Shengwei Xiong et al. investigates the usage of 3D CT reconstruction technique for preoperative planning, navigation during surgery as well as localization of uretero-pelvine stricture. The authors were able to show that 3D Models increased intraoperative stone clearance rate, reduced surgical complications and recurrence of obstruction and stones. Alltogeter patiens seem to profit from 3D preplaning.

The study is easy to follow and read and clearly structured. The topic is relevant for urologists and urological surgeons.

I have some minor remarks:

- some references are more than 10years old, I think the study would benefit from literature which is up to date

- an abbreviations section is missing - please include

- for the discussion section - which other medical fields or for which other medical conditions/surgerys is 3D modelling/preplaning used/ applyed or investigated - a comparrison would be interesting

Author Response

Dear reviewers,

On behalf of my co-authors, we thank you very much for giving us a chance to revise our article. Those comments are all valuable and very helpful for revising and improving our paper, as well as provide important guiding significance to our study.

We have read your comments carefully and have made revision, which marked in red in the paper. We have tried our best to revise our manuscript according to the comments. Attached please find the revised version, which we would like to submit for your kind consideration. Here, we have written a point-by-point response as following:

Comments and Suggestions for Authors:

This interesting single-center study by Shengwei Xiong et al. investigates the usage of 3D CT reconstruction technique for preoperative planning, navigation during surgery as well as localization of uretero-pelvine stricture. The authors were able to show that 3D Models increased intraoperative stone clearance rate, reduced surgical complications and recurrence of obstruction and stones. Altogether patients seem to profit from 3D preplaning.

The study is easy to follow and read and clearly structured. The topic is relevant for urologists and urological surgeons.

I have some minor remarks:

- some references are more than 10 years old, I think the study would benefit from literature which is up to date

Answer: Thanks for your suggestion. We deleted some old references and replaced them with some recent ones, which were marked in red.

- an abbreviations section is missing - please include

Answer: Thanks for your advice. We added the abbreviations section above the “References” section (marked in red).

- for the discussion section - which other medical fields or for which other medical conditions/surgery is 3D modelling/preplaning used/ applyed or investigated - a comparison would be interesting.

Answer: Your suggestions are highly appreciated. We have added the description of 3D reconstruction technology for the surgical planning and navigation of prostate cancer and complex renal tumor. And the superiority of 3D reconstruction technology for complex renal stone diseases, compared with prostate cancer and kidney tumors, was also added. Please see the first two paragraphs (marked in red) of the Discussion section.

We appreciate for your warm work earnestly, and hope that the correction will meet with approval.

Yours Sincerely,

Xuesong Li MD

Reviewer 2 Report

Authors should be congratulated for their work. Obstructive UPJO is a pediatric concern with high prevalence in the general population as the Authors presented in the discussion. The manuscript is well written. The tables are easily readable and well-presented, but several points warrant a mention:

- Are data available on the placement of the stent during the procedure? And the removal after the procedure? Are data available on the presence of hydronephrosis after the stent removal if it was placed?

- Are available data on the renal functioning of these 26 patients pre and post-treatment from the renal scintigraphic imaging?

- Was the follow-up performed with the same methods for every one of these patients? How was it conducted?

- What was the approach for the remotion of renal calculi? Were the patients studied from a metabolic point of view? 

- Are data available on the complications of the patients after LP? Are they evaluated with the Calvin Lindo classification?

- Are trends of eGFR and creatinine available from the pre-operative day up to the post-operative days after surgery? Are any data available on the blood loss during the procedures and on the Hb?  Some recent articles evaluated the trend of Hb for the robotic approach to prostatectomy, delineating a line to follow in the management of prostate cancer patients. It would be interesting to study if UPJO patients could benefit from evidence-based studies, improving their tailored-management such as previous Authors demonstrated (PMID: 36363477).

Author Response

Dear reviewers,

On behalf of my co-authors, we thank you very much for giving us a chance to revise our article. Those comments are all valuable and very helpful for revising and improving our paper, as well as provide important guiding significance to our study.

We have read your comments carefully and have made revision, which marked in red in the paper. We have tried our best to revise our manuscript according to the comments. Attached please find the revised version, which we would like to submit for your kind consideration. Here, we have written a point-by-point response as following:

Comments and Suggestions for Authors

Authors should be congratulated for their work. Obstructive UPJO is a pediatric concern with high prevalence in the general population as the Authors presented in the discussion. The manuscript is well written. The tables are easily readable and well-presented, but several points warrant a mention:

- Are data available on the placement of the stent during the procedure? And the removal after the procedure? Are data available on the presence of hydronephrosis after the stent removal if it was placed?

Answer: Thank you for your advice. We briefly described the process of placement of the stent during the procedure in Figure 4. We used a 7F double-J stent, and we placed the stent after partial anastomosis between the renal pelvis and ureter was completed. The double-J stent was removed 2 to 3 months after surgery, which was briefly described in the second paragraph of “Perioperative and follow-up data” section (marked in red). We didn't evaluate the presence of hydronephrosis immediately after removal of the stent, but we monitored the patients’ symptoms. If the patient has a recurrence of back pain, we will conduct further examination and evaluation. Only 1 patient recurred back pain 16 months after surgery in our study, which was confirmed caused by renal stones.

- Are available data on the renal functioning of these 26 patients pre and post-treatment from the renal scintigraphic imaging?

Answer: Thanks for your suggestion. Less than half of our patients performed the renal scintigraphic imaging before the operation, so we didn’t show the related data in this manuscript. We evaluated the patients’ renal function mainly by monitoring the changes in serum creatinine and eGFR, and postoperative CTU, 3D CT models and dynamic MRI monitoring are also helpful.

- Was the follow-up performed with the same methods for every one of these patients? How was it conducted?

Answer: Thank you very much for your suggestion. The follow-up of all patients was performed with the same method. The postoperative follow-up was conducted by the surgical team through outpatient clinic at regular intervals (every 3-6 months) after the operation. We have a standardized follow-up team, and relevant preoperative and postoperative data were recorded in our multicenter RECUTTER (Reconstruction of Urinary Tract: Technology, Epidemi-ology and Result) database (http://www.3dmi.com.cn/login/pkufh). Postoperative renal function was evaluated by monitoring the changes in serum creatinine and estimated glomerular filtration rate. Ultrasonography and CT urography or 3D CT reconstruction technology were used as follow-up examinations to evaluate postoperative hydronephrosis and residual stones. The above details were described in “Patient selection and management” section (marked in red).

- What was the approach for the remotion of renal calculi? Were the patients studied from a metabolic point of view?

Answer: Thank you very much for your advice. In our study, for patients with small renal calculi located deep in the renal calyx, only TLP or RALP was performed, renal calculi were not removed. For patients with renal calculi of bigger sectional area, deep position and more involved renal calyces (Figure 2B and Figure 2C), TLP or RALP combined with pyelolithotomy using laparoscopic forceps was performed. For patients with more complicated and larger renal calculi or staghorn calculi, the stones were extracted ureteroscope or cystoscope, and sometimes holmium laser lithotripsy was introduced. The above details were described in “Patient grouping and surgical planning” section (marked in red) and “Surgical techniques” section. In this study, we failed to analyze the renal stones from a metabolic point of view, as we focused on evaluating the effect of 3D reconstruction technology on preoperative planning and intraoperative navigation of patients with UPJO and concomitant renal calculi.

- Are data available on the complications of the patients after LP? Are they evaluated with the Calvin Lindo classification?

Answer: Thanks for your comment. We evaluated the intraoperative and postoperative complications using the Calvin-Lindo classification. Only one patient with endoscopic lithotomy developed Clavien–Dindo grade 3 complication. And this patient had a displaced ureteral stent 2 days after surgery, which was subsequently replaced with a new one. The remaining patients did not develop Clavien–Dindo grade 2 or higher complications during perioperative period. The above details were described in the second paragraph of“Perioperative and follow-up data / Results” section (marked in red) and Table 2.

- Are trends of eGFR and creatinine available from the pre-operative day up to the post-operative days after surgery? Are any data available on the blood loss during the procedures and on the Hb?  Some recent articles evaluated the trend of Hb for the robotic approach to prostatectomy, delineating a line to follow in the management of prostate cancer patients. It would be interesting to study if UPJO patients could benefit from evidence-based studies, improving their tailored-management such as previous Authors demonstrated (PMID: 36363477).

Answer: Your suggestions are highly appreciated. In our study, almost all patients have only one blood draw after surgery to test renal function. Therefore, we collected the renal function results (eGFR and creatinine) of patients before surgery, 1 day, 3 months and 12 months after surgery to observe the changing trends. The results were showed in Figure 5. We could observe the trend of increased eGFR and decreased creatinine on the average levels after surgery, although these changes were not statistically significant. As our patients had little intraoperative blood loss and no significant changes in postoperative hemoglobin levels, we didn’t show the data of blood loss and hemoglobin levels. The above details were described in the first paragraph of“Perioperative and follow-up data / Results” section (marked in red) and Figure 5.

We appreciate for your warm work earnestly, and hope that the correction will meet with approval.

Yours Sincerely,

Xuesong Li MD

Round 2

Reviewer 2 Report

The authors had exhaustively answered the suggestion. Now the article is suitable for publication.